# The Influence of Cationic Nitrosyl Iron Complex with Penicillamine Ligands on Model Membranes, Membrane-Bound Enzymes and Lipid Peroxidation

**DOI:** 10.3390/membranes12111088

**Published:** 2022-11-02

**Authors:** Darya A. Poletaeva, Yuliya V. Soldatova, Anastasiya V. Smolina, Maxim A. Savushkin, Elena N. Klimanova, Nataliya A. Sanina, Irina I. Faingold

**Affiliations:** 1Federal Research Center of Problems of Chemical Physics and Medicinal Chemistry, Russian Academy of Sciences, Academician Semenov Avenue, 1142432 Chernogolovka, Russia; 2Faculty of Fundamental Physical and Chemical Engineering, Moscow State University, 1142432 Moscow, Russia; 3Medicinal Chemistry Research and Education Center, Moscow Region State University, 1142432 Mytishchy, Russia

**Keywords:** tetranitrosyl iron complex, nitric oxide, liposomes, lipid bilayer, cell membrane, biomimetic membrane, free radical, lipid peroxidation, membrane-bound enzymes, monoamine oxidase, glycation, cardiovascular diseases

## Abstract

This paper shows the biological effects of cationic binuclear tetranitrosyl iron complex with penicillamine ligands (TNIC–PA). Interaction with a model membrane was assessed using a fluorescent probes technique. Antioxidant activity was studied using a thiobarbituric acid reactive species assay (TBARS) and a chemiluminescence assay. The catalytic activity of monoamine oxidase (MAO) was determined by measuring liberation of ammonia. Antiglycation activity was determined fluometrically by thermal glycation of albumine by D-glucose. The higher values of Stern–Volmer constants (K_SV_) obtained for the pyrene located in hydrophobic regions (3.9 × 10^4^ M^−1^) compared to K_SV_ obtained for eosin Y located in the polar headgroup region (0.9 × 10^4^ M^−1^) confirms that TNIC–PA molecules prefer to be located in the hydrophobic acyl chain region, close to the glycerol group of lipid molecules. TNIC–PA effectively inhibited the process of spontaneous lipid peroxidation, due to additive contributions from releasing NO and penicillamine ligand (IC50 = 21.4 µM) and quenched luminol chemiluminescence (IC50 = 3.6 μM). High activity of TNIC–PA in both tests allows us to assume a significant role of its radical-scavenging activity in the realization of antioxidant activity. It was shown that TNIC–PA (50–1000 μM) selectively inhibits the membrane-bound enzyme MAO-A, a major source of ROS in the heart. In addition, TNIC–PA is an effective inhibitor of non-enzymatic protein glycation. Thus, the evaluated biological effects of TNIC–PA open up the possibility of its practical application in chemotherapy for socially significant diseases, especially cardiovascular diseases.

## 1. Introduction

Membranes are essential for life because they enable organisms to maintain homeostasis, providing a consistent environment inside the cell. Biological membranes consist of lipid bilayers as basic structure units and membrane proteins, which perform many specific vital functions. Phosphatidylcholine (PC) is the most abundant phospholipid in all mammalian cells. Lipid molecules spontaneously form bilayers in aqueous environments. Moreover, a biological membrane is a two-dimensional liquid in which the structural units are free to move laterally [1]. Membrane fluidity is crucial to many membrane functions.

Lipid membranes can largely influence membrane proteins’ local structure, dynamics, and activity [2]. Membrane proteins mediate processes that are vital for cells. Membrane-embedded transporters uptake vital substances and remove unwanted ones across the cell membranes, receptors bind to signaling molecules and initiate a physiological response, and membrane-embedded enzymes catalyze chemical reactions. Membrane proteins represent more than 60% of the current drug targets [3]. Furthermore, in many cases, chemical or functional changes in membranes are central to the pathogenesis of a disease.

Water-soluble dinitrosyl iron complexes with thiol-containing ligands, being the “working form” of nitric oxide (NO), are important regulators of physiological processes in living organisms, such as inhibition of platelet aggregation, dilation of vascular smooth muscles via activating soluble guanylate cyclase, and enhancement of cardiac resistance to ischemia [4]. Synthetic analogs of these natural NO carriers are a promising base for the design of drugs to treat a wide range of health problems, including cardiovascular disease, cancer, and others [5,6,7,8].

The present work describes the study of the biological effects of a DNIC dimer, cationic binuclear tetranitrosyl iron complex with naturally occurring thioligand–penicillamine (TNIC–PA), [Fe_2_(S(C(CH_3_)_2_CH(NH_3_)COOH))_2_(NO)_4_]SO_4_⋅5H_2_O (Figure 1). Binuclear sulfur-nitrosyl iron complexes in water and physiological solutions spontaneously generate NO without additional activation, forming dinitrosyl mononuclear intermediates (DNIC) in solutions. The time dependence of the amount of released NO during the decomposition of the TNIC–PA in aerobic aqueous solutions has previously been shown [9]. It was found that 500 s after dissolution of TNIC–PA, the NO amount in the solution was ~60 nM, which was twice as much as for another complexes. The major decomposition products within 11 min under aerobic conditions are NO and penicillamine ligands, and one of the ligands is protonated.

This complex is one of the leaders in this class of synthetic low-molecular-weight nitrosyl iron complexes in terms of its ability to influence vascular tone. TNIC–PA decreased blood pressure upon bolus injection into the aorta of an isolated perfused Wistar rat heart. The vasodilatory efficacy of TNIC–PA was higher than that of sodium nitroprusside (a direct acting vasodilator) at similar concentration of 3 μM. An intravenous bolus injection of TNIC–PA resulted in a significant dose-dependent reduction of the systolic blood pressure in anesthetized Wistar rats, in the same manner as nitroglycerin [10]. Thus, previous studies have shown that TNIC–PA is promising for NO therapy of acute coronary syndrome.

We have previously shown that anionic tetranitrosyl iron complex with thiosulfate ligands (TNIC) has a pronounced antioxidant activity and acts as a free radical scavenger, and also inhibits the catalytic activity of the mitochondrial enzymes cytochrome c oxidase and monoamine oxidase A [12]. Hence, this work aimed to study the antioxidant properties of the TNIC–PA complex and its effects on the activity of mitochondrial enzymes and protein glycation because it has enhanced NO donating ability; that is, 500 s after the dissolution of complexes TNIC and TNIC–PA, the amounts of generated NO in the solutions were ~34, and ~55 nM, respectively [9].

We aim to study TNIC–PA interaction with the membrane of PC liposomes using a fluorescent probes technique and its effect on lipid peroxidation and outer mitochondrial membrane enzymes (monoamine oxidase) activity.

It is known that drug effectiveness strongly correlates with the distribution coefficient, which is a measure of the amount of drug that will enter into and/or through a lipid membrane. Distribution coefficients are usually estimated for drugs by partitioning the ionized species in the oil phase. Some studies have shown that liposomes are a better alternative than traditional methods used to estimate distribution coefficients because those methods cannot account for the ionic interaction between drugs and lipids, particularly when drugs are charged, as in the case of the studied cationic complex [13]. We have used fluorescent probes with known locations in the liposomal membrane (eosin Y and pyrene) to study TNIC–PA distribution into membrane and the resulting changes in membrane fluidity.

The oxidation of membrane phospholipids and other polyunsaturated fatty acids (PUFA) is one of the most prominent manifestations of oxidative stress. The potential role of oxidative damage in cardiovascular diseases have been studied continuously for several decades. Reactive oxygen species (ROS) and other oxidants can cause the oxidation of lipids, proteins, and DNA, resulting in tissue damage [14]. The modulation of lipid peroxidation may be an important strategy for slowing the onset and the progression of a number of socially significant diseases, such as cardiovascular diseases [15]. The main primary products of lipid peroxidation are lipid hydroperoxides (LOOH) [16].

The main source of cellular reactive oxygen species (ROS) is mitochondria, and membrane-bound mitochondrial enzymes monoamine oxidase (MAO) is one of the major mitochondrial sources of oxidative stress in cardiomyocytes [17]. Additionally, the heart is a target organ for oxidative stress-related injuries, in which cardiomyocyte regeneration is insufficient [18]. All of these factors makes the heart vulnerable to the accumulation of lipid peroxidation products formed as a result of oxidative damage. In light of the established roles of ROS in heart diseases, MAO inhibitors are now actively being investigated for the treatment of cardiac dysfunction.

In addition, it is interesting to study how TNIC–PA can affect the process of non-enzymatic protein glycation. Advanced glycation end products (AGEs) are compounds formed by non-enzymatic reactions between free amino groups of proteins and carbonyl groups of reducing sugars. This process is involved in the complications associated with several disorders including diabetes, cardiovascular disease and cancer [19].

Taken together, targeting lipid peroxidation, MAO, and non-enzymatic protein glycation may be a good strategy to look for compounds with inherent therapeutic potential to treat cardiovascular diseases.

## 2. Materials and Methods

### 2.1. Chemicals and Materials

Eosin Y, pyrene, PC from egg yolk, aminoguanidine, sodium azide, serotonin, benzylamine, Nessler’s reagent, thiobarbituric acid (TBA), trichloroacetic acid (TCA), penicilamine (PA), tert-butyl hydroperoxide (TBHP) and luminol were purchased from Sigma (Merck Life Science LLC, Moscow, Russian Federation). Bovine serum albumin (BSA) (fraction V) and D-glucose were purchased from Life Science (USA). Folin’s reagent was purchased from Applichem (AppliChem GmbH, Darmstadt, Germany).

### 2.2. Investigated Compound

TNIC–PA: cationic binuclear complex with penicillamine ligands [Fe_2_(SC(CH_3_)_2_CH(NH_3_)COOH)_2_(NO)_4_]SO_4_⋅5H_2_O. TNIC–PA was synthesized using previously reported methods [11]. 

NO was generated from sodium nitrite mixed with ferrous chloride by adding hydrochloric acid. The gas was passed through two traps filled with sodium hydroxide and collected in a glass gasometer. Saturated NO solution (2.1 mM) in water was prepared at 20 °C by bubbling purified NO for 30 min through deoxygenated water in a septum-capped vial. The solvent was deoxygenated with Ar for 45 min before bubbling with NO.

The operation procedure in the argon atmosphere was described in [20].

### 2.3. Liposomes Study

#### 2.3.1. Vesicle Preparation

Egg PC was dissolved in ethanol and placed in a round-bottomed flask. The solvent was removed using a rotary evaporator Heidolph Hei-Vap (Heidolph Instruments GmbH & Co, Schwabach, Germany). After drying under vacuum for 1 h, the residual lipid film was hydrated with Tris buffer (pH = 7.2) and vortex-mixed. Small unilamellar vesicles were obtained by sonicating the multilamellar vesicles for 10 min to clarity on ice under an argon atmosphere using a Bandelin SONOPULS HD3100 homogenizer (BANDELIN electronics GmbH & Co, Berlin, Germany). Metal debris was removed by centrifugation.

#### 2.3.2. Dynamic Light Scattering

The size distribution of liposomes was determined by dynamic light scattering (DLS) using a Photocor Compact particle size analyzer (Photocor Instruments, Tallinn, Estonia) (TEC stabilized diode laser 654 nm; scattering angle was set to 90°). DLS measurements were performed after microporous filtration (pore size 0.45 μM). The experimental data were analyzed using DynaLS v. 2.8.3 software. The hydrodynamic radius value (R_h_) of vesicles was determined using the Stokes–Einstein equation: D = kT⁄(6πηR_h_), where D is the diffusion coefficient, k is the Boltzmann’s constant, T is the absolute temperature, and η is the dynamic viscosity. The study shows lipid suspension mainly composed of single unilamellar liposomes with R_h_ ~ 12.5 nm.

#### 2.3.3. Fluorescence Measurements

Fluorescence measurements were taken using a Cary Eclipse fluorescence spectrometer (Agilent, Santa Clara, CA, USA). The sample compartment was maintained at a constant temperature with the use of circulating water through the cell holder. A liposomes suspension (2 mL; 0.5 mM PC) containing eosin Y (2.0 μM), or pyrene (5.0 μM) was titrated by successive additions of TNIC–PA solution (1 mM). The fluorescence quenching of fluorescent probes with TNIC–PA was recorded at 350–600 nm. The excitation and emission slit width was adjusted to 5 nm, and the excitation wavelength was selected as 517 nm for eosin Y and 337 nm for pyrene. All fluorescence titration experiments were performed manually with a 1 mL microsyringe. Dimer: monomer ratios were calculated by comparing the fluorescence intensity at 475 nm to that at 394 nm, using 337 nm as the excitation wavelength. The figures showing fluorescence quenching spectra were made using Origin 7.5.

### 2.4. Tissue Preparation

The work was carried out in accordance with EU Directive 2010/63/EU. Hybrid BDF1 mice (about 6 months old) were killed by decapitation. For the MAO and TBA reactive substances (TBARS) assays, each brain was rapidly excised, frozen in liquid nitrogen, and stored at 80 °C until use. For the luminol chemiluminescence assay, the freshly prepared brains were thawed and homogenized using a WisdWiseTis HG-15D homogenizer (Daihan Scientific Group, Wonju, South Korea) for 2 min in a buffer (0.1 M Tris-HCl, pH 7.4). Protein concentrations were determined by the Lowry method [21].

### 2.5. Luminol Chemiluminescence Assay

Luminol-dependent chemiluminescence produced by mouse brain homogenates was measured using a 1250 Luminometer (LKBWallac, Turku, Finland) according to the assay by Di Meo et al. [22] with minor modifications. The reaction mixture consisted of mouse brain homogenate (protein concentration 0.1 mg/mL) in 0.1 M Tris-HCl (Sigma-Aldrich, St. Louis, MO, USA), pH 7.4, luminol (Sigma-Aldrich, St. Louis, MO, USA) (0.05 mM), TBHP (Sigma-Aldrich, St. Louis, MO, USA) (0.073 M) and tested compounds. This was placed in a light-proof luminometer chamber, and the background reading was recorded. All measurements were carried out at 37 °C. The gain control was set to give a reading of 10 mV using a built-in standard. The free radical content in mouse brain homogenate was evaluated by the change in the light sum (the area under the kinetic curve of luminescence intensity for the entire luminescence time). All compounds were examined at least in triplicate. The effects of tested compounds were measured as the percentage decrease in the light sum relative to the control (a sample with no compound added).

### 2.6. TBARS Assay

The procedure was performed in accordance with the method of Ohkawa et al. [23] with minor modifications. Briefly, mouse brain homogenate (1:5 (*w*/*v*) ratio of brain tissue to 0.1 M K-Na-phosphate buffer (pH 7.2)) was incubated for 30 min at 37 °C with the tested compound, and the reaction was terminated by the addition of 0.4 mL of 17% (*w*/*v*) TCA (Sigma-Aldrich, St. Louis, MO, USA). Following centrifugation for 20 min at 1300 g, 0.5 mL of 0.8% (*w*/*v*) of TBA (Sigma-Aldrich, St. Louis, MO, USA) was added to 1 mL of supernatant, heated for 30 min at 95 °C, and then cooled to room temperature. The optical density of the TBARS, which corresponds to the produced malondialdehyde (MDA), was measured at 532 nm against a blank using an Agilent Cary 60 UV-Vis spectrophotometer (Agilent, Santa Clara, CA, USA). The MDA concentration was calculated using an extinction coefficient of 1.56 × 10^5^ M^−1^ cm^−1^. Compounds were tested in a concentration range of 8.5–670 µM. All compounds were examined at least in triplicate. IC_50_ values represent the concentration that caused 50% inhibition of LP.

### 2.7. MAO Activity Assay

MAO-A and B activity in mouse brain homogenate was assessed according to a protocol described earlier [24,25]. Mouse brain homogenate was prepared in a schuett homgen glass–teflon semi-automatic homogenizer (Schuett-biotec GmbH, Göttingen, Germany) in a 1:5 (*w*/*v*) ratio of brain tissue to K-Na-phosphate buffer (pH 7.2, 0.1 M). The method is based on a spectrophotometric measurement of the amount of ammonia evolved during enzymatic deamination of benzylamine (catalyzed by MAO-B enzyme) and serotonin (catalyzed by MAO-A). Protein concentration was determined by the Lowry method [21].

### 2.8. The Antiglycation Assay

The antiglycation assay was performed according to the methods described in [26] with slight modifications. The sample contained 0.5 ml BSA (4 mg/mL), 0.4 mL D-glucose (0.4 M) in Na-phosphate buffer (pH = 7.4, sodium azide content 0.02%) and 0.1 ml of the test compound. The final assay volume was 1 mL. Buffer was added to control samples instead of compounds. Aminoguanidine was used as a positive control [27]. The reaction was allowed to proceed at 60 °C for 48 h. The reaction was stopped by adding 10 μl of 100% (*w*/*v*) TCA. The TCA-added mixture was kept at 4 °C for 10 min before centrifugation at 135,000 rpm using an Ohaus Frontier 5515R centrifuge (4 min, 4 °C). The precipitate was redissolved with 1 ml alkaline PBS (pH = 10).

The relative amount of glycated BSA was immediately determined based on fluorescence intensity using a Cary Eclipse spectrofluorometer (Agilent, Santa Clara, CA, USA). The excitation and emission wavelengths used were λ_exc/em_ = 370/440 nm for vesperlysines-like AGE and λ_exc/em_ = 335/385 for pentosidine-like AGE [28]. The percentage of AGE formation was calculated as follows: AGEs (%) = (fluorescence intensity (sample) − fluorescence intensity (blank of sample))/(fluorescence intensity (control)−fluorescence intensity (blank of control)) × 100. The IC_50_ (concentration of half-maximal inhibition of the process) was calculated.

### 2.9. Acute Toxicity Evaluation in Mice

To determine acute toxicity, TNIC–PA was administered to animals intragastrically and intraperitoneally, once, at doses of 500–800 mg/kg and 75–250 mg/kg, respectively, according to the method described in [29]. Male hybrids of the BDF1 line weighing 20–22 g (6 mice in each group) were used. Physiological saline solution was used for control injections. For intragastric administration of TNIC–PA solutions, a specialized atraumatic intragastric tube was used. Mortality of the animals and the clinical presentation of intoxication were observed for 14 days. The lethal dose estimate (LD_50_) (the dose that kills 50% of the test animal population) was used as a significant parameter in measuring acute toxicity.

### 2.10. Statistical Analysis

All analyses were carried out using the statistical software packages Origin 9.1 and GraphPad Prism 8. All in vitro assays for the compound were performed in triplicate, and values were represented as mean ± standard error of the mean (SEM). The significance between individual groups was analyzed by using Student’s *t*-test. *p* values < 0.05 were considered statistically significant.

## 3. Results and Discussion

### 3.1. Fluorescent Probe Studies in Model Membrane

TNIC–PA is an effective NO donor and could be a promising drug candidate [9]. The interaction of biologically active compounds with the cell membrane plays a major role in their pharmacokinetics, since they must penetrate one or more phospholipid bilayers to reach the intracellular targets and exhibit pharmacological activity, so drug–membrane interactions are inevitable.

The direct interaction of TNIC–PA with the membrane can be studied using powerful and useful model systems such as phospholipid liposomes that mimic a natural bilayer lipid membrane. In the present study, we used small unilamellar PC liposomes. There are three possible types of interaction between TNIC–PA and lipids, interaction with the hydrophilic head groups, with hydrophobic tails, and both simultaneously.

In the present work, we studied the interaction of the TNIC–PA with hydrophilic and hydrophobic regions of model membranes using a fluorescent probe method. The goal was to understand the interactions of TNIC–PA with phospholipids and, in particular, to assess the level of spontaneous insertion of TNIC–PA into the lipid bilayer.

The use of fluorescence probes is a sensitive and quick technique that gives meaningful information about the environment of the fluorescent chromophore in membranes. 

The xanthene dye eosin Y was found to be located in the polar headgroup region of the bilayer, 17–19 Å from the bilayer center [30]. Analysis of fluorescence quenching of eosin Y binding to zwitterionic PC model membranes by TNIC–PA can give essential information about TNIC–PA interaction with the hydrophilic region of the lipid bilayer.

As shown in Figure 2, the addition of TNIC–PA steadily decreased the fluorescence intensity of eosin Y in liposomes solution without any changes in the spectrum, displaying the classic fluorescence quenching effect.

Figure 3 shows the Stern–Volmer plot of the quenching experiment. The data could be linearly fitted adequately, yielding a value for the Stern–Volmer constant of K_SV_ 0.9 × 10^4^ M^−1^. The same measurements were taken in Tris-HCl buffer (pH = 7.2) to compare obtained K_SV_ values. In an aqueous solution, the value of the constant was three times lower, at 0.3 × 10^4^ M^−1^. The results indicate that TNIC–PA has a binding affinity for polar head groups of PC model membranes.

Pyrene was the fluorescent probe of choice to study the distribution of TNIC–PA into the hydrophobic regions of the lipid membrane. Pyrene has excellent solubility in cell and model membranes and is localized in the hydrocarbon core. Furthermore, due to rapid lateral diffusion, pyrene molecules can form excited-state dimers (excimers) [31]. The addition of TNIC–PA complex to the liposomes suspension resulted in effective pyrene fluorescence quenching and increased the excimer to monomer (I’/I) ratio (Figure 4 and Figure 5).

The value for the Stern–Volmer constant was K_SV_ = 3.9×10^4^ M^−1^. Since excimer formation requires some freedom of movement of the pyrene molecules during the lifetime of the excited state (80–120 ns in liposomes), the increasing I’/I ratio could be due to a decrease in the viscosity of the microenvironment of the pyrene [32]. This confirms TNIC–PA molecules’ distribution into the lipid bilayer’s hydrophobic regions.

It is known that the lipid bilayer’s physical state affects the lipid oxidation rate in liposomes [33]. A range of therapeutic antioxidants, such as lazaroids [34,35,36], and α-tocopherol [37], selectively bind to certain domains of the lipid bilayer, provoking changes in the molecular packaging and dynamics of membranes. It has been suggested that the nature and site of the drug–bilayer interaction may play an essential role in the effectiveness of these compounds as antioxidants in lipid peroxidation (LPO). Free radicals formed in the body disrupt the cell membranes’ structure, leading to the development of various pathological conditions [38]. However, the significance of oxidative stress in pathology development is very variable, since the effectiveness of antioxidant cellular defence can be limited in some diseases. Therefore, low molecular weight compounds—radical scavengers (traps)—can play an important role in preventing the damaging effects of reactive oxygen species.

### 3.2. Antioxidant Activity

The antioxidant and free radical scavenging activity were determined by using two standard methods, chemiluminescence (CL) and the TBARS method. The results are shown in Figure 6a,b.

#### 3.2.1. Radical Scavenging Capacity in Luminol Chemiluminescence Assay

The ability of TNIC–PA to inhibit lipid peroxidation and exhibit antiradical activity was studied by luminol chemiluminescence assay. This method is based on antioxidant-dependent quenching of chemiluminescence generated from lipids of mice brain homogenate [22]. Luminol was used to sensitively and selectively detect hydrogen peroxide (H_2_O_2_), hydroxyl radicals (OH), hypochlorite (ClO^−^), peroxynitrite (ONOO^−^), and lipid peroxyl radicals [22].

It was found that TNIC–PA reduces the intensity of luminol chemiluminescence in a wide range of concentrations (1–100 μM). The dose-response curve is shown in Figure 6a. The IC_50_ value for TNIC–PA was determined as 3.6 ± 0.9 μM. As can be seen, TNIC–PA is several times more effective than standard antioxidant BHT (IC_50_ = 73.1 ± 4.9 μM). Such a pronounced effect suggested a great antiradical capacity of TNIC–PA.

NO did not affect CL intensity at concentrations of 10–500 μM (*p* > 0.05). PA is known to be capable of inhibiting the production of active oxygen species by human leucocytes (which was shown using luminol-dependent chemiluminescence) [39]. Therefore, we assume that the antiradical activity of TNIC–PA is closely linked to the presence of penicillamine ligands.

#### 3.2.2. TBARS Assay

Antioxidant activity of TNIC–PA was also measured using a TBARS assay. Results showed that TNIC–PA can slow down the accumulation of MDA during the breakdown of unsaturated fatty acids in membranes. The IC_50_ value was determined as 21.4 ± 0.6 µM.

As shown in the kinetic curves in Figure 6b, there is a significant difference in MDA accumulation in the presence of the three tested compounds. The addition of 0.5 mM of TNIC–PA produced a potent inhibition of MDA accumulation in mice brain homogenate, as did the addition of NO (the concentration was twice as high, 1 mM), whereas the inhibition was significantly lower in the case of PA. We suggest that TNIC–PA antioxidant activity is due to the additive effect of both penicillamine ligand and NO released during the degradation of the complex.

Thus, the results have shown the remarkable antioxidant capacity of TNIC–PA in spontaneous and initiated lipid peroxidation, which is closely linked to the TNIC–PA interaction with the membrane, especially with nonpolar (hydrophobic) regions.

Polar lipids, structural components of cell membranes, may control the physiological state of membrane organelles (e.g. mitochondria). During the process of lipid peroxidation, free radicals attack polyunsaturated fatty acids, triggering the formation of lipid peroxyl radicals. This imbalance between the production and accumulation of ROS in cells causes oxidative stress [16]. One of the primary sources of endogenous ROS production is the mitochondria [40], through various mechanisms including enzymatic reactions. Recently, MAOs have been considered an essential source of ROS. MAOs are localized in the outer membrane of mitochondria and exist as two different isoforms, MAO-A and MAO-B. MAOs catalyze the oxidative deamination of biogenic amines, generating H_2_O_2_ and aldehydes as byproducts.

Several studies have shown that MAO-A plays a key role in regulating physiological cardiac function and in the development of acute and chronic heart diseases through the regulation of substrate concentration and intracellular production of ROS [41,42,43]. Thus, MAO-A is an important factor in cardiac ageing that can be a target for drugs exerting cardioprotective actions [44].

### 3.3. Determination of the Inhibitory Effect of TNIC–PA on Membrane-Bound Enzymes MAO-A and B

One of the study’s goals was to evaluate the potential of TNIC–PA, which has a potent affinity to membranes, to inhibit the MAO enzymes. TNIC–PA and PA were evaluated in vitro against MAO-A and B in mice brain homogenate. The data obtained for TNIC–PA are shown in Figure 7.

Residual MAO activities in the presence of different TNIC–PA concentrations were determined. TNIC–PA demonstrated potent MAO-A inhibitory activities. The obtained values of percentage inhibition ranged from 33% to 71%. MAO-B activity did not change in the presence of TNIC–PA (5–1000 μM). PA (5–1000 μM) was ineffective at inhibiting both MAOs. Since the PA does not affect MAOs, selective MAO-A inhibition by TNIC–PA can be determined by the complex’s NO-donor activity or by the presence of biologically active thiols, which can liberate flavin from monoamine oxidase [45].

Serotonin (5-hydroxytryptamine, 5-HT) is a preferential substrate of MAO-A, involved in a number of functions in peripheral tissues. The location of 5-HT in the membrane is stabilized by hydrogen bonds between its hydroxyl group and lipid headgroups and allows 5-HT to scavenge ROS, preventing membrane oxidation [46]. Therefore, MAO-A inhibitors might be useful in managing the outcome of tissue damage associated with oxidative stress, and the antioxidant capacity of TNIC–PA might also manifest itself through the inhibition of MAO-A.

### 3.4. Antiglycating Activity

The effect of TNIC–PA on the process of non-enzymatic glycation of BSA in vitro was studied in this work. The tested compound was shown to be an effective inhibitor of the glycation process. We observe a decrease in AGE fluorescence at wavelengths λ_exc/em_ = 370/440 nm, corresponding to the fluorescence of vesperlysines-like AGEs, and λ_exc/em_ = 335/385 nm, corresponding to pentosidine-like AGEs. The dose/response curves are shown in Figure 8.

IC_50_ (the concentration able to achieve 50% inhibition of AGE formation) for the formation of vesperlysines-like AGEs was determined to be 7.7 ± 0.9 µM, and for pentosidine-like AGEs to be 4.3 ± 0.4 µM. At the same time, the penicillamine had no effect on the non-enzymatic glycation of albumin in the concentration range from 0.1 to 500 µM. TNIC–PA is a more effective glycation inhibitor than aminoguanidine (IC_50_ for forming vesperlysines-like AGEs of 1294 μM), which was used in in vitro experiments as a positive control [47,48,49].

Even though TNICs possess a broad spectrum of biological activities, there were no data regarding their effects on the process of non-enzymatic glycation. Inhibition of AGE formation by a compound from the class of synthetic low-molecular-weight TNICs is reported here for the first time and is found to have promising potential for further research in the field of cardiovascular diseases and diabetes mellitus.

The processes of non-enzymatic glycation and lipid peroxidation are closely related. Oxidative stress induces endogenous formation and accumulation of both products of lipid peroxidation (such as 4-hydroxy-2-nonenal and malonic dialdehyde) and AGEs, which can be produced from the same precursors such as glyoxal and methylglyoxal [50]. Thus, lipid peroxidation inevitably leads to the accumulation of AGEs and vice versa. The structure, fluidity, and functional integrity of membranes are disturbed not only by lipid peroxidation but also by the glycation of membrane proteins and the action of AGEs [51]. Excessive AGEs and dicarbonyl compounds have detrimental effects on the permeability of mitochondrial membranes and mitochondrial respiratory function [52,53].

Binuclear sulfur-nitrosyl iron complexes in water and physiological solutions spontaneously generate NO without additional activation. NO is known to be able to inhibit cellular lipid peroxidation and it can be an effective antioxidant in vivo because only very low levels are required [54]. NO can terminate chain reactions in lipid peroxidation, due to interaction with lipid radicals (L^•^), lipid peroxy radicals (LOO^•^) and lipid alkoxyl radicals (LO^•^) [55,56]. As an antioxidant, TNIC–PA may prevent AGE formation by blocking autoxidative and glycoxidation pathways [57]. The mechanisms of the antioxidant and antiglycation activity of TNIC–PA are shown in Figure 9.

### 3.5. Acute Toxicity Assessment

In the study of acute toxicity in BDF1 mice, it was found that, with a single intraperitoneal (intragastric) administration, TNIC–PA is a moderately toxic substance. The LD_50_ value was 181.5 ± 10.8 mg/kg (581.1 ± 17.9 mg/kg).

## 4. Conclusions

This work shows the high ability of TNIC–PA to bind liposomal membranes. TNIC-PA prefers to be located in the hydrophobic acyl chain region, close to the glycerol group of lipid molecules, which was proved by higher values of the K_SV_ constant obtained for the pyrene, compared to K_SV_ obtained for eosin Y.

We confirm the effect of TNIC–PA as an inhibitor of lipid peroxidation, due to additive contributions from releasing NO and penicillamine ligand (IC_50_ = 21.4 ± 0.6 µM). The antioxidant mechanisms of TNIC–PA are based on its ability to scavenge hydroxyl radicals, hypochlorite, peroxynitrite, and eliminate hydrogen peroxide, which have been proven by chemiluminescence (IC_50_ = 3.6 ± 0.9 μM).

It has been shown that TNIC–PA selectively inhibits membrane-bound enzyme MAO-A in the concentration range of 50–1000 μM. Penicillamine does not affect the activity of MAO. MAO-A inhibition is probably determined by the direct interaction of releasing NO with the enzyme molecule or by the presence of biologically active thiols, which can liberate flavin from monoamine oxidase. In addition, TNIC–PA is an effective inhibitor of the process of non-enzymatic protein glycation in vitro.

Oxidative stress is highly implicated in cardiovascular diseases such as myocardial infarction, ischemia/reperfusion, and heart failure. The main source of cellular ROS is mitochondria, and monoamine oxidases are one of the major mitochondrial sources of oxidative stress in cardiomyocytes. In addition, complications associated with cardiovascular diseases are closely related to protein glycation. The ability of TNIC–PA to modulate these target proteins and pathways opens up the possibility of its practical application in chemotherapy for socially significant diseases, especially cardiovascular diseases.

## Figures and Tables

**Figure 1 membranes-12-01088-f001:**
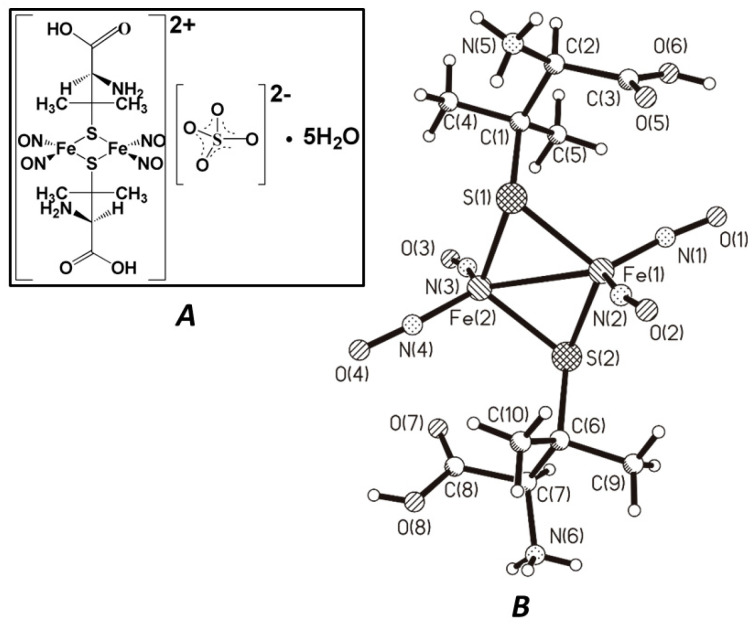
The TNIC–PA formula (**A**) and the structure of its dication [Fe_2_(S(C(CH_3_)_2_CH(NH_3_)COOH))_2_(NO)_4_]^2+^ (**B**) (from X-ray data [11]).

**Figure 2 membranes-12-01088-f002:**
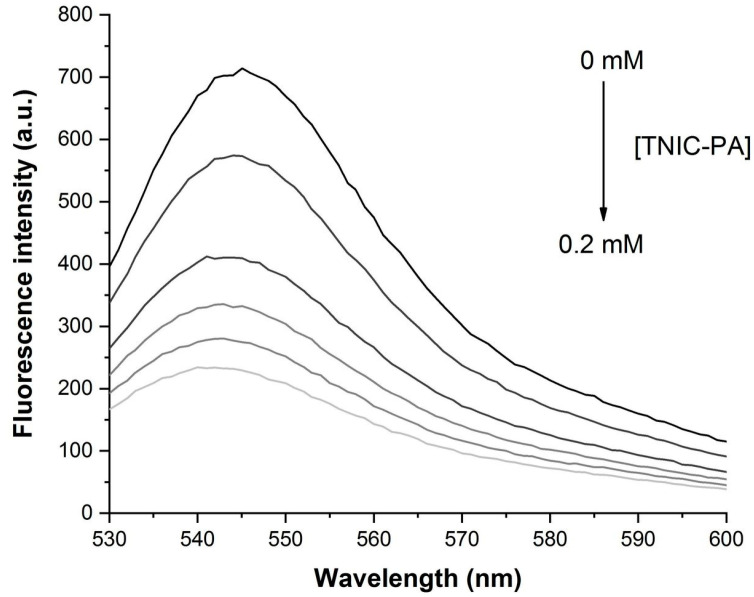
Fluorescence quenching of eosin Y in the presence of TNIC–PA in liposomes suspension. Fluorescence intensity measurements are taken after each incremental addition of TNIC–PA up to the maximum molarity of 0.2 mM.

**Figure 3 membranes-12-01088-f003:**
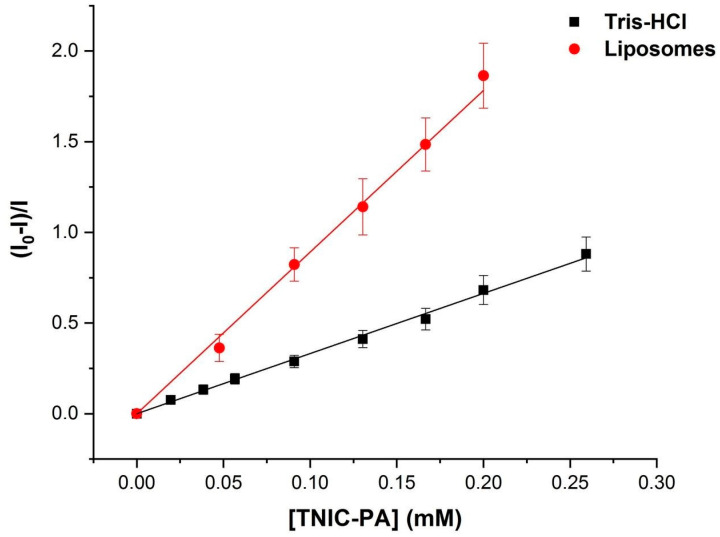
Stern–Volmer plot for the fluorescence quenching of eosin Y in the presence of TNIC–PA.

**Figure 4 membranes-12-01088-f004:**
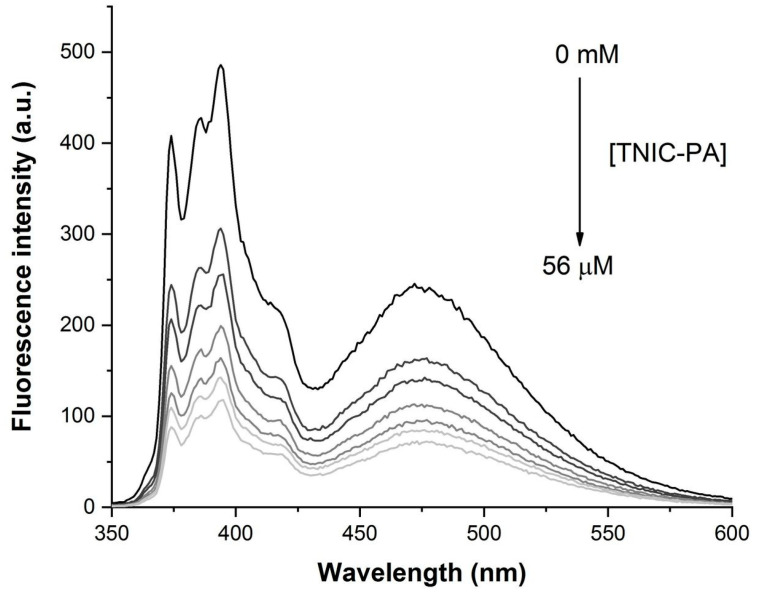
Fluorescence quenching of pyrene in the presence of TNIC–PA in liposomes suspension. Fluorescence intensity measurements are taken after each incremental addition of TNIC–PA up to the maximum molarity of 56 µM.

**Figure 5 membranes-12-01088-f005:**
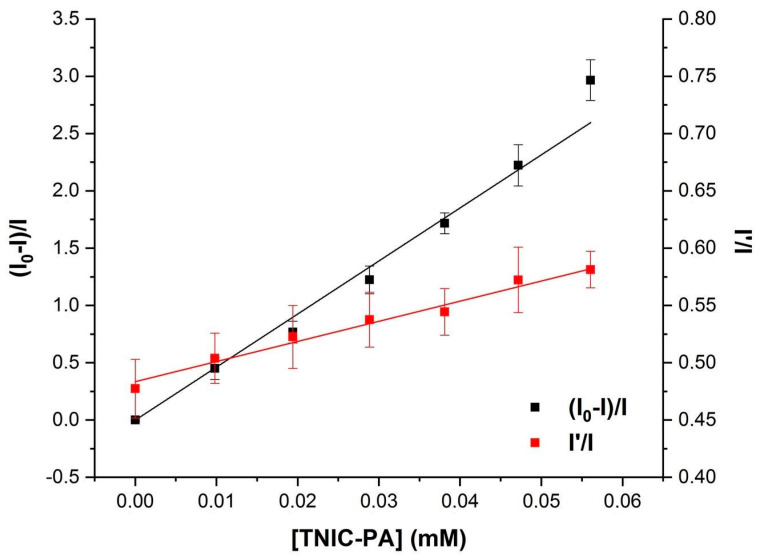
Stern–Volmer plot for the fluorescence quenching of pyrene in the presence of TNIC–PA (black) and the fluorescence ratio of excimer (I’) over monomer (I) as a function of TNIC–PA concentration (red).

**Figure 6 membranes-12-01088-f006:**
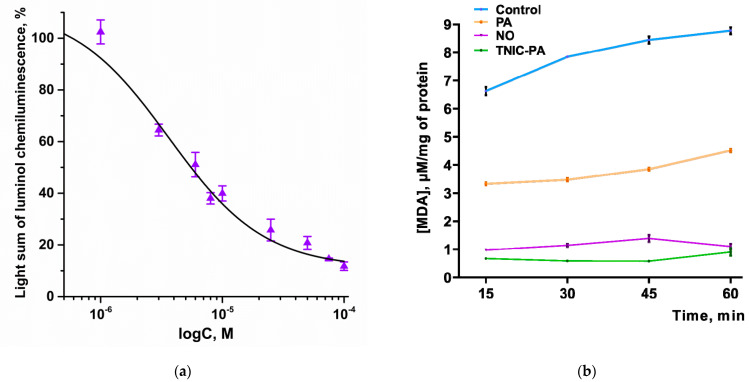
(**a**) Dose-effect curve of the dependence of the light sum of chemiluminescence of luminol in the presence of TNIC–PA on the concentration of TNIC–PA. CL data are presented as a percentage relative to the control. (**b**) The rate of MDA accumulation in the mouse brain homogenate under the action of TNIC–PA [0.5 mM], NO [1 mM], and PA [0.5 mM]. Control sample–mouse brain homogenate free of the test compounds. vs. control (Student’s *t*-test).

**Figure 7 membranes-12-01088-f007:**
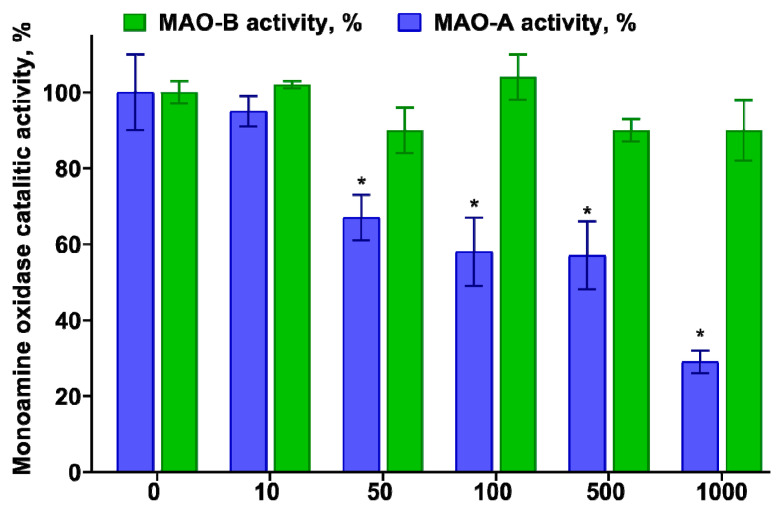
The percentage of MAO inhibition by TNIC–PA. Control—MAO assay performed without any inhibitor shows the maximum enzyme’s catalytic activity. * *p* < 0.05 vs. Control.

**Figure 8 membranes-12-01088-f008:**
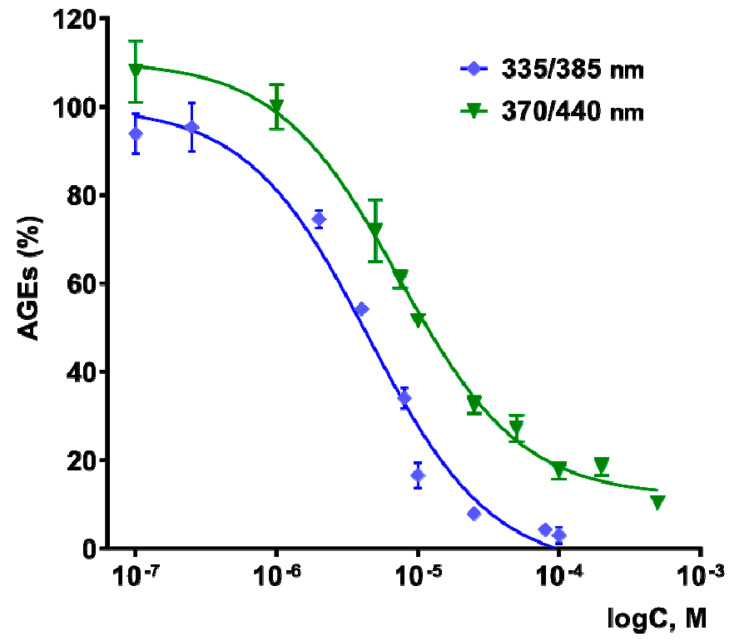
Dose-effect curves for vesperlysines-like AGEs (green) and pentosidine-like AGEs (blue) formation in the presence of TNIC–PA. λ_exc/em_ = 370/440 nm for vesperlysines-like AGEs and λ_exc/em_ = 335/385 nm for pentosidine-like AGEs.

**Figure 9 membranes-12-01088-f009:**
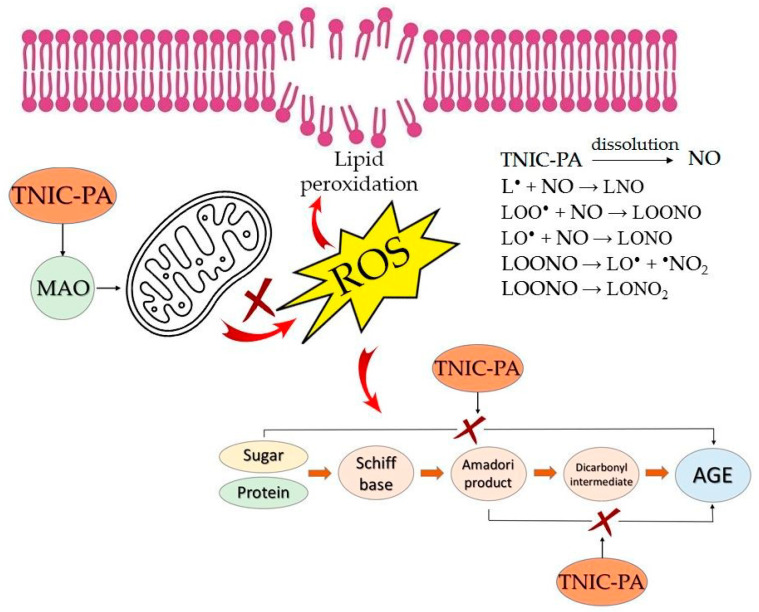
Schematic showing the mechanisms of antioxidant and antiglycation activity of TNIC–PA.

## Data Availability

Not applicable.

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
