# Peer review of "The Influence of Cationic Nitrosyl Iron Complex with Penicillamine Ligands on Model Membranes, Membrane-Bound Enzymes and Lipid Peroxidation"

_membranes, 2022, doi:10.3390/membranes12111088_

Round 1

Reviewer 1 Report

This paper described biological effects of cationic binuclear tetranitrosyl iron complex with penicillamine ligands (TNIC-PA). Several assay methods have been performed to demonstrate the pharmacological effects of the compounds, and no discrepancies regarding these experiments have been found. However, the novelty of this study could not be understood at all; there is too little information on TNIC-PA. What pharmacological activity has been reported for this compound? Also, have the assay methods used in this study been performed in the past? Furthermore, we do not know why the assay method performed in this study was selected out of all the basic assay methods in the world to demonstrate various pharmacological activities. It is stated in the paper that the assay method is important, and I am aware of this, but this is a general statement, and the reason why the assay method was selected should be stated in relation to the chemical structure of TNIC-PA and its existing pharmacological activity.

Author Response

We would like to thank the reviewers for their thoughtful comments and efforts towards improving our manuscript. Please see the attachment.

Manuscript ID: membranes-1970851

Reviewer comments:

Point 1 This paper described biological effects of cationic binuclear tetranitrosyl iron complex with penicillamine ligands (TNIC-PA). Several assay methods have been performed to demonstrate the pharmacological effects of the compounds, and no discrepancies regarding these experiments have been found. However, the novelty of this study could not be understood at all; there is too little information on TNIC-PA. What pharmacological activity has been reported for this compound?

Response 1 Thank you for your comments. We agree that the novelty of our study is not well explained and too little information about the studied compound TNIC-PA was provided. Introduction was revised; information about TNIC-PA itself and its previously studied pharmacological activity was added.

Point 2 Also, have the assay methods used in this study been performed in the past? Furthermore, we do not know why the assay method performed in this study was selected out of all the basic assay methods in the world to demonstrate various pharmacological activities. It is stated in the paper that the assay method is important, and I am aware of this, but this is a general statement, and the reason why the assay method was selected should be stated in relation to the chemical structure of TNIC-PA and its existing pharmacological activity.

Response 2 The methods used in this study are accurate, reliable and well known. These methods were chosen in order to show the effect of the TNIC-PA on biological targets, which together may indicate the effectiveness of the complex as a potential drug for the treatment of cardiovascular diseases. To make it clear, we have revised the introduction. It should also be said, that some methods have been previously used to study the biological activity of anionic tetranitrosyl iron complex with thiosulfate ligands [Faingold, I.I., Kotelnikova, R.A., Smolina, A.V. et al. Antioxidant Activity of Tetranitrosyl Iron Complex with Thiosulfate Ligands and Its Effect on Catalytic Activity of Mitochondrial Enzymes In vitro. Dokl Biochem Biophys 488, 342–345 (2019). https://doi.org/10.1134/S1607672919050120 and I.I. Faingold, D.A. Poletaeva, Yu.V. Soldatova, A.V. Smolina, O.V. Pokidova, A.V. Kulikov, N.A. Sanina, R.A. Kotelnikova. Effects of albumin-bound nitrosyl iron complex with thiosulfate ligands on lipid peroxidation and activities of mitochondrial enzymes in vitro, Nitric Oxide, Volume 117, 2021, Pages 46-52, https://doi.org/10.1016/j.niox.2021.10.002]. Activity of molecules is reflected in their chemical structure indeed. Hence, similar molecules have similar activities. Chemical structure of TNIC-PA as NO-donor with naturally occurring thioligand – penicillamine was the basis of our choice of research methods.

Reviewer 2 Report

The investigators described the membrane interaction and the antioxidant and anti-glycating activities of TNIC-PA. Each assay was well organized, and the results showed potent activity of TNIC-PA. It would be desirable to correlate and discuss the results of the assays presented separately. From my point of view, the reviewer raised the following points:

1.Line 232, the investigators described “TNIC-PA is an effective NO donor and could be a promising drug candidate.” Please cite the appropriate references. In addition, investigators should explain why they focused on TNIC-PA among the DNIC dimers.

2. Line 283-284, the investigators described “This confirms TNIC-PA molecules' distribution into the lipid bilayer's hydrophobic regions.” The reviewer agrees that their experimental results likely support this description. However, the reviewer wondered why and how the hydrophilic molecular structure of TNIC-PA with charged functional groups can arrive at the hydrophobic moiety of the bilayer membranes. Can investigators add experimental data by using other compounds as positive and negative controls? Another option is to estimate membrane partition coefficient or a kind of LogP value of TNIC-PA.

3. Line 430-432. The investigators concluded “Combining the evaluated biological effects of TNIC-PA opens up the possibility of its practical application in chemotherapy of socially significant diseases, especially cardiovascular diseases.” This description has not been fully understood by the reviewer. Please add more sentences to explain this description to persuade readers.

4. The author should add one figure about the possible molecular mechanism(s) of TNIC-PA to express antioxidant activity and inhibition of glycation.

Minor point

The investigators should also show chemical structure of TNIC-PA in Figure 1. 

Author Response

We would like to thank the reviewers for their thoughtful comments and efforts towards improving our manuscript. Please see the attachment.

Manuscript ID: membranes-1970851

Reviewer comments: The investigators described the membrane interaction and the antioxidant and anti-glycating activities of TNIC-PA. Each assay was well organized, and the results showed potent activity of TNIC-PA. It would be desirable to correlate and discuss the results of the assays presented separately. From my point of view, the reviewer raised the following points:

Point 1 Line 232, the investigators described “TNIC-PA is an effective NO donor and could be a promising drug candidate”. Please cite the appropriate references. In addition, investigators should explain why they focused on TNIC-PA among the DNIC dimers.

Response 1 Thank you for your comments. The citation about NO-donor activity of TNIC-PA has been added [Boris L. Psikha, Elena A. Saratovskikh, Ilya V. Sulimenkov, Alina S. Konyukhova, Natalia A. Sanina. Reactions of water-soluble binuclear tetranitrosyl iron complexes of the μ-S structural type with adenosine triphosphoric acid: Kinetics and reaction mechanism, Inorganica Chimica Acta, Volume 531, 2022, 120709, https://doi.org/10.1016/j.ica.2021.120709] (reference 9).The reason why we focused on TNIC-PA is its enhanced NO donating ability and previously studied pharmacological activities. All information was added to the Introduction section.

Point 2 Line 283-284, the investigators described “This confirms TNIC-PA molecules' distribution into the lipid bilayer's hydrophobic regions.” The reviewer agrees that their experimental results likely support this description. However, the reviewer wondered why and how the hydrophilic molecular structure of TNIC-PA with charged functional groups can arrive at the hydrophobic moiety of the bilayer membranes. Can investigators add experimental data by using other compounds as positive and negative controls? Another option is to estimate membrane partition coefficient or a kind of LogP value of TNIC-PA.

Response 2 The TNIC-PA molecule is indeed hydrophilic and readily dissociates in proton media into the dication [Fe2(S(C(CH3)2CH(NH3)COOH))2(NO)4]2+ and the dianion SO42-. We assume that the decomposition products of the dication can interact with the hydrophobic region of lipid bilayer. Previously, we have particularly studied the properties of the TNIC-PA in aqueous media and analyzed its interaction with the albumin [Olesya V. Pokidova, Alexandra Yu. Kormukhina, Alexander I. Kotelnikov , Tatyana N. Rudneva, Konstantin A. Lyssenko, Natalia A. Sanina, Features of the decomposition of cationic nitrosyl iron complexes with N-ethylthiourea and penicillamine ligands in the presence of albumin, Inorganica Chimica Acta 524 (2021) 120453 https://doi.org/10.1016/j.ica.2021.120453]. It was found that the original UV-visible spectrum of TNIC-PA has pronounced bands at 310 and 360 nm, which is consistent with the literature data on binuclear complexes [A.F. Vanin, A.P. Poltorakov, V.D. Mikoyan, L.N. Kubrina, D.S. Burbaev, Nitric Oxide. 23 (2010) 136–149, https://doi.org/10.1016/j.niox.2010.05.285]. In addition, at a concentration of the complex of 0.6∙mM, experimental bands were observed in the region of 430–440 nm, which, most likely, refer to d → d excitations, with a participation of the sulfur orbitals [M. Jaworska, Z. Stasicka, New J. Chem. 29 (2005) 604–612, https://doi.org/10.1039/b409519g]. It has been shown that the optical density of TNIC-PA decrease with time, indicating its gradual decay. This decomposition, as shown in the work, is accompanied by the release of NO, penicillamine, and various highly reactive iron-containing nitrosyl intermediates [Boris L. Psikha, Elena A. Saratovskikh, Ilya V. Sulimenkov, Alina S. Konyukhova, Natalia A. Sanina. Reactions of water-soluble binuclear tetranitrosyl iron complexes of the μ-S structural type with adenosine triphosphoric acid: Kinetics and reaction mechanism, Inorganica Chimica Acta, 531 (2022) 120709 https://doi.org/10.1016/j.ica.2021.120709]. Such intermediates are able to interact with the hydrophobic part of albumin, which was shown in our previous work using similar complex with thiourea, for which the landing sites of nitrosyl intermediates were found by molecular docking, including in the hydrophobic pocket of albumin [Pokidova O.V., Luzhkov V.B., Emel'yanova N.S., Krapivin V.B., Kotelnikov A.I., Sanina N.A., Aldoshin S.M. Effect of albumin on the transformation of dinitrosyl iron complexes with thiourea ligands. Dalton Transactions (2020) 49(36), 12674-12685 https://doi.org/10.1039/D0DT02452J]. We assume that TNIC-PA can interact in a similar way with a hydrophobic region of lipid bilayer, but the detailed mechanism of this interaction, of course, will be of interest for our further research using quantum chemical calculations.

It was impossible to estimate distribution coefficient for TNIC-PA experimentally because of its rapid decomposition upon dissolution. Therefore, the only possibility here is to calculate the Log P for the ligand (penicillamine) using ALOGPS 2.1 software. Lop P (octanol/water) for TNIC-PA ligand penicillamine is -1.7.

Point 3 Line 430-432. The investigators concluded “Combining the evaluated biological effects of TNIC-PA opens up the possibility of its practical application in chemotherapy of socially significant diseases, especially cardiovascular diseases.” This description has not been fully understood by the reviewer. Please add more sentences to explain this description to persuade readers.

Response 3 To make it more clear, we revised the conclusion section.

Point 4 The author should add one figure about the possible molecular mechanism(s) of TNIC-PA to express antioxidant activity and inhibition of glycation.

Response 4 The figure with possible mechanisms of antioxidant and antiglycation activity of TNIC-PA was added.

Minor point The investigators should also show chemical structure of TNIC-PA in Figure 1.

Response Chemical structure of TNIC-PA was added to Figure 1.

Round 2

Reviewer 1 Report

The authors have revised the manuscript based on the review comments. I have determined that the paper is acceptable for publication.